# Climate Change and Habitat Fragmentation: Implications for the Future Distribution and Assisted Migration of *Kobresia pygmaea*

**DOI:** 10.3390/plants14233585

**Published:** 2025-11-24

**Authors:** Zongcheng Cai, Fayi Li, Shancun Bao, Hairong Zhang, Jianjun Shi

**Affiliations:** 1Academy of Animal Science and Veterinary, Qinghai University, Xining 810016, China; ys230951310630@qhu.edu.cn (Z.C.); yb240909000126@qhu.edu.cn (F.L.);; 2Key Laboratory of Alpine Grassland Ecology in the Three-River-Source Region, Ministry of Education, Xining 810016, China; 3Qinghai Provincial Key Laboratory of Adaptive Management of Alpine Grasslands, Ministry of Education, Xining 810016, China

**Keywords:** Tibetan Plateau, *Kobresia pygmaea*, MaxEnt, suitable habitat, climate change impact

## Abstract

Understanding alpine plants’ survival and reproduction is crucial for their conservation in climate change. This study, based on 273 valid distribution points, utilizes the MaxEnt model to predict the potential habitat and distribution dynamics of *Kobresia pygmaea* under both current and future climate scenarios (SSP126, SSP245, SSP370, SSP585), while clarifying the key factors that influence its distribution. The study indicates that elevation (3527.99–6054.54 m) is the dominant factor influencing its distribution. The current suitable habitat is primarily concentrated in southern and central Tibet, northwestern Sichuan, and southern Qinghai on the Tibetan Plateau, with a total area of 1.13 × 10^5^ km^2^, of which high- and moderate-suitability areas account for 1.76 × 10^4^ km^2^ and 3.2 × 10^4^ km^2^, respectively. Under future climate scenarios (2050s–2070s), the overall distribution pattern remains concentrated on the Tibetan Plateau, but the suitable area exhibits a trend of initial expansion followed by contraction. By the 2050s, the total suitable area increases across all scenarios, with the most pronounced expansion under SSP126. By the 2070s, however, the total suitable area decreases under high-emission scenarios, declining by 9.50% under SSP370 and 6.76% under SSP585, respectively. The reduction in high-suitability areas is more severe, with a maximum decline of 58.75% under SSP3-7.0. Dynamic change analysis shows that approximately 70% of the current high-suitability areas remain stable by the 2050s, with range expansion occurring under low-emission scenarios toward southeastern Tibet, northwestern Sichuan, and southern Golog in Qinghai. In contrast, habitat contraction intensifies by the 2070s, particularly under the SSP5-8.5 scenario, where the reduced area reaches 1.6 times the current high-suitability extent. Centroid shift analysis indicates that the distribution center of suitable habitats migrates northward or northeastward, with a maximum displacement of 206.51 km under the SSP1-2.6 scenario by the 2050s. The results suggest that short-term climate warming may alleviate low-temperature constraints, facilitating the upward and poleward expansion of *Kobresia pygmaea* into higher-elevation areas. However, prolonged and intensified warming will likely lead to degradation of core habitats, posing a significant threat to its long-term persistence. This study provides a scientific basis for the conservation of alpine ecosystems on the Tibetan Plateau and for developing adaptive management strategies under climate change.

## 1. Introduction

Global climate change has substantially influenced biodiversity and altered species distribution patterns [1,2]. Li found that in *Leontopodium nanum*, a species native to the Qinghai–Tibet Plateau, the suitable habitat is projected to contract by 50% under the high-emission scenario (SSP585). Concurrently, new suitable habitats emerge in the central-western and northern regions of the plateau, accounting for 71.86% of the current distribution, indicating significant range expansion. The centroid of the suitable habitat shifts northward by 23.94–342.42 km, reflecting persistent geographical displacement. This co-occurring pattern of “contraction–expansion–displacement” represents a typical ecological response of alpine plants to climate warming through spatial redistribution [3]. As a consequence, more and more species are experiencing habitat enlargement, shrinkage, and geographic displacement [4,5]. The impacts of climate change on ecosystem dynamics are especially evident in alpine grassland ecosystems, which represent highly fragile regions [6,7]. In alpine regions, due to their unique geographical environment, species require narrow ecological amplitudes and specific growth requirements for survival and reproduction. Coupled with rapid climate change in recent years, the survival of these species is facing increasing threats [8]. Climate change not only exerts direct impacts on species habitats, but also enhances ecosystem vulnerability by altering ecological processes and disrupting competitive interactions among species within biological communities [9,10]. Therefore, studying the impacts of global climate change on species diversity, particularly the distribution patterns of alpine plants in high-altitude and cold regions, has become a major focus in contemporary ecological research.

Geographic information science, as an emerging technology, has been widely applied in ecology, particularly through the use of ecological niche models (ENMs) and species distribution models (SDMs), to predict the potential distribution areas of plant species [11,12]. This technology plays a crucial role in crop breeding, conservation of rare and endangered plant species, and the management and prevention of plant diseases and pests [13,14]. Currently, MaxEnt, Bioclim, and GARP are among the most commonly used species distribution models (SDMs) for predicting suitable habitats. However, under conditions of limited sample size, ease of operation, and high prediction accuracy, the MaxEnt model has emerged as the preferred choice and has gained widespread recognition and application in studies of species distribution [15,16]. MaxEnt is a recently introduced technique that is less influenced by extraneous parameters and is capable of achieving high-precision predictions in modeling species’ potential distributions [11,15]. In summary, ecological niche models (ENMs) and species distribution models (SDMs) integrate geographic information systems (GIS), climatic variables, and environmental data to construct mathematical models for predicting the potential suitable habitats of plant species [17,18].

*Kobresia pygmaea* (*Cyperaceae*, genus *Kobresia*) is a perennial cushion-forming herbaceous plant characterized by its dwarf stature, linear leaves, and well-developed root system [19]. Its morphological configuration exhibits strong adaptation to alpine environments, possessing desirable traits such as tolerance to low temperatures, drought, trampling, and soil erosion [20]. It is a dominant and constructive species in alpine grasslands. *K. pygmaea* possesses significant economic, social, and ecological functions, plays a crucial role in the alpine regions of the Qinghai–Tibet Plateau, and has received widespread attention. However, these habitat conditions are highly susceptible to the impacts of climate change and human activities, leading to an increasingly fragmented and marginal distribution pattern of *K. pygmaea* in natural environments. For instance, studies have demonstrated that grassland degradation on the Qinghai–Tibet Plateau is driven by the synergistic effects of anthropogenic and natural factors. Anthropogenic activities—such as overgrazing, excessive cultivation, unregulated harvesting of medicinal plants, mineral exploitation, infrastructure construction, and tourism development—have substantially altered ecosystem structure and function. Natural drivers, including climate change (e.g., rising temperatures and altered precipitation regimes), bioturbation by wildlife, and pest outbreaks, further exacerbate ecosystem instability. The combined impact of these stressors has led to widespread habitat loss and increased landscape fragmentation across the plateau region [21]. Currently, many studies focus on the photosynthetic characteristics, cold resistance mechanisms, and leaf traits of *K. pygmaea* [20]. However, research on its predicted distribution under future climate scenarios remains limited. Therefore, investigating the impacts of climate change on the distribution of *K. pygmaea* and predicting the shifts in its suitable habitats under future climate scenarios have practical significance for the long-term conservation of this species.

Researchers utilize the four climate scenarios and emission pathways proposed by the Intergovernmental Panel on Climate Change (IPCC)—SSP126, SSP245, SSP370, and SSP585—to simulate future climate change conditions and improve the accuracy of species distribution predictions [22,23]. These four scenarios represent different emission pathways characterized by varying concentrations of greenhouse gases and changes in mean temperature, providing critical insights for comprehensively understanding and conserving alpine *K. pygmaea* [24]. For example, Zhang predicted the current and future potential habitats of Paeoniaceae using four climate scenarios [25]; Li employed the MaxEnt model to project the distribution shifts of three herbaceous *Coptis* species in China under climate change [26].

This study, based on the distribution patterns of *K. pygmaea* on the Qinghai–Tibet Plateau, employs the MaxEnt model to simulate its potential suitable habitats, analyzes the dominant factors influencing its spatial distribution, and predicts changes in its distribution patterns under different future climate scenarios. This not only helps to elucidate the potential threats of climate change to this alpine species but also provides scientific guidance for its conservation and ecological management.

## 2. Materials and Methods

### 2.1. Sources of Species Distribution Data

This study compiled a total of 480 occurrence records of *K*. *pygmaea* by integrating data from the Global Biodiversity Information Facility (GBIF) (https://www.gbif.org/citation-guidelines, accessed on 21 July 2025), the National Plant Specimen Resource Center, published literature, and field surveys. To avoid spatial autocorrelation and overfitting, spatial thinning was performed using the ENMTools (V4.06.00) package with a 5 km resolution grid, retaining one occurrence per cell. A total of 273 unique occurrence points were ultimately retained for suitable habitat prediction (Figure 1).

### 2.2. Environment Variable Data and Screening

The meteorological and topographic data used in this study, including 19 bioclimatic variables and one elevation variable, were obtained from the WorldClim database (version 2.1, ~1 km resolution), representing current conditions and future projections for the 2050s and 2070s under four Shared Socioeconomic Pathways (SSPs): SSP126, SSP245, SSP370, and SSP585 [27]. Soil and elevation data were sourced from HWSD v1.2, with future conditions assumed unchanged due to a lack of projections [28]. All environmental data is raster data.

### 2.3. MaxEnt Model Optimization and Parameter Setting

This study used the MaxEnt model to predict the potentially suitable distribution area of *K. pygmaea*, and the technical route is shown in Figure 2.

The potential distribution of *K. pygmaea* was predicted using the MaxEnt model, with hyperparameters optimized via the kuenm package (RM = 2). Occurrence data and environmental variables were input with a 25% random test fraction, linear and quadratic feature types, and 10 replicates; logistic output was generated, and variable importance was assessed using jackknife analysis (Figure 3).

### 2.4. Suitable Area Division and Area Calculation

The MaxEnt-predicted potential habitats were reclassified in ArcGIS into four suitability classes: unsuitable (0–0.2), low (0.2–0.4), medium (0.4–0.6), and high (0.6–1.0). Pixel counts within each class were calculated, converted to proportions of the total, and multiplied by China’s total land area to estimate the actual area of each suitability zone.

### 2.5. Temporal and Spatial Evolution of Suitable Areas

Future suitability rasters were imported into ArcGIS and converted to binary rasters using a 0.6 threshold. Expansion, contraction, and centroid shifts of the suitable habitat for *K. humilis* under climate change scenarios were analyzed, with changes in area and centroid displacement distance quantified.

## 3. Results and Analysis

### 3.1. Species Distribution Sites and Filtering Results of Environmental Variables

The 273 species, along with 19 environmental variables (including 18 soil variables and 1 elevation variable), were imported into MaxEnt. However, some of the climate variables provided by WorldClim may exhibit strong correlations. To avoid potential multicollinearity issues that could affect the reliability of jackknife tests, Pearson correlation analysis was conducted to assess the relationships among environmental variables (Figure 4). After filtering, 37 environmental variables were reduced to 18 variables with higher contribution rates, comprising 11 climatic variables, 6 soil variables, and 1 elevation variable (Table 1).

### 3.2. MaxEnt Accuracy Evaluation

The area under the curve (AUC) value of the receiver operation characteristic (ROC) output by the MaxEnt model was used to evaluate the accuracy of prediction results. The closer the AUC value is to 1, the more accurate the prediction. Typically, AUC < 0.7 indicates poor prediction, 0.7 ≤ AUC < 0.8 indicates fair prediction, 0.8 ≤ AUC < 0.9 indicates good prediction, and 0.9 ≤ AUC < 1.0 indicates excellent prediction.

During the model calibration phase, a total of 24 candidate models were developed. These models were generated using different combinations of feature classes (linear, quadratic, product, hinge, and threshold) and regularization multipliers (RM, ranging from 0.5 to 4.0 in increments of 0.5). Model performance was systematically evaluated to identify the optimal configuration. The best-performing model was selected based on the lowest AICc (corrected Akaike Information Criterion) value, which corresponds to an RM of 1.5 with the LQPHT feature combination (Linear, Quadratic, Product, Hinge, Threshold). This model achieved a training AUC value of 0.970, indicating excellent discriminatory power (Figure 5b). Furthermore, the average omission rate on the independent validation dataset was 7.3% (based on the 10% training presence logistic threshold), which is below commonly accepted thresholds and further supports the model’s high predictive reliability (Figure 5a).

### 3.3. Key Environmental Variables Affecting the Distribution of K. pygmaea

MaxEnt was used for the knife-cutting analysis of environmental variables to determine the important ecological variables affecting *K. pygmaea*. As shown in Figure 6, when only one variable was used, the gain value (blue band) of altitude (wc.2.1_2.5m_elev) was the largest, indicating that this variables themselves contained more useful in-formation about the distribution of *K. pygmaea*. At the same time, among other variables, altitude (wc.2.1_2.5m_elev) decreased the most, indicating that this variable contained more information about the distribution of *K. pygmaea* that other variables did not. Therefore, altitude were the most important environmental variables affecting the distribution of *K. pygmaea*. According to the results of the response curve of leading environmental variables (Figure 6), when the probability of suitable growth of *K. pygmaea* is greater than 0.6 (high suitability area), the altitude suitable for distribution is 3527.99 m~6054.54 m (Figure 7).

### 3.4. Suitable Areas for K. pygmaea Under Current Climate Conditions

Under current climatic conditions, the suitable habitats of *K. pygmaea* in China are mainly distributed in the central and southern regions of the Tibetan Plateau, the central and southern parts of Qinghai Province, and the northwestern regions of Sichuan Province (Figure 8). The global suitable habitat area totals 1.13 × 10^5^ km^2^, accounting for 1.18% of the total land area of the country (Figure 9). Among them, the areas of high and moderate suitable habitats are 1.76 × 10^4^ km^2^ and 3.2 × 10^4^ km^2^, respectively, accounting for 15.59% and 28.44% of the total suitable habitat area. Under current climatic conditions, the high suitability areas for *K. pygmaea* are primarily concentrated in southern and eastern Tibet, as well as northwestern Sichuan. Medium suitability areas are mainly distributed in central Tibet, parts of eastern Tibet, and southern Qinghai. Low suitability areas are widespread across the Qinghai–Tibet Plateau, with the exception of the Qaidam Basin in Qinghai, which exhibits unsuitable or marginal habitat conditions for this species.

### 3.5. Suitable Area of K. pygmaea in Different Climate Scenarios in the Future

Under projected climate scenarios for the 2050s and 2070s, the potential suitable habitat of *Kobresia pygmaea* remains largely consistent with its current distribution, primarily concentrated in the Qinghai–Tibet Plateau region (Figure 10 and Figure 11). From the 2050s to the 2070s, as greenhouse gas concentrations increase, the total suitable habitat area of *K. pygmaea* exhibits a trend of initially increasing followed by a subsequent decline. Under different climate scenarios in the 2050s, the total suitable habitat area of *K. pygmaea* is larger than the current extent. By the 2070s, however, the total suitable area gradually decreases to varying degrees, with particularly notable contractions under SSP370 (1.02 × 10^5^ km^2^ and SSP585 (1.05 × 10^5^ km^2^), showing reductions of 9.50% and 6.76%, respectively, compared to the present-day distribution. Moreover, the high suitability area of *K. pygmaea* under both the 2050s and 2070s climate scenarios is generally smaller than the current extent, exhibiting an overall trend of initial increase followed by decline. The maximum high suitability area is projected to occur under the SSP12.6 scenario in the 2050s (1.95 × 10^4^ km^2^), representing an 11.08% increase compared to the current high suitability area (1.76 × 10^4^ km^2^). By the 2070s, the high suitability area of *K. pygmaea* continues to decline, reaching its minimum under the SSP370 scenario (7.24 × 10^3^ km^2^), which represents a 58.75% reduction compared to the current high suitability area. This indicates that, by the 2070s, increasing greenhouse gas concentrations will lead to a progressive contraction of suitable habitats for *K. pygmaea*, particularly under high-emission pathways.

### 3.6. Trends in the Gain and Loss of Suitable Habitat Area for K. pygmaea Under Future Climate Change Scenarios

Under future climate scenarios, the high suitability area of *K. pygmaea* undergoes significant changes, with the most pronounced shifts projected for the 2070s (Figure 11).

By the 2050s, approximately 70% of the current suitable habitat area is projected to remain stable, while the newly gained area accounts for 20.38% of the current high suitability area, and the contracted area comprises 48.97% of the current high suitability area. Under the low emission climate scenario (SSP126), the expansion areas are primarily concentrated in southern and parts of eastern Tibet, northern Sichuan (particularly in parts of Aba Prefecture), and southern parts of Golog Prefecture in Qinghai Province. With increasing greenhouse gas concentrations, the expansion area is minimal and the contraction area is maximal under the SSP585 climate scenario.

By the 2070s, the newly gained high suitability area expands to 16.17% of the current high suitability area, while the contracted area increases to 1.6 times the extent of the current area. The contraction areas are primarily concentrated in southern and eastern Tibet, as well as in parts of Aba Prefecture in northern Sichuan. With increasing greenhouse gas concentrations, the extent of contraction exhibits a trend of initially decreasing and then increasing, reaching its maximum under the high-emission scenario (SSP585), where the high suitability habitat is projected to shrink by 25.95% compared to the current area.

### 3.7. Centroid Shift Trends of K. pygmaea Under Future Climate Scenarios

Based on the geometric center of the suitable habitat of *K. pygmaea*, the centroid shifts under different climate scenarios in the 2050s and 2070s were analyzed (Figure 12). The results indicate that, under different climate scenarios, the migration direction and distance of *K. pygmaea* are primarily concentrated within the Qinghai–Tibet Plateau region. Under the SSP126 scenario in the 2050s, the most significant migration occurs, with the centroid of *K. pygmaea* shifting 206.51 km northeast. The second-largest shift is projected under the SSP370 scenario in the 2050s, with a 110.57 km displacement toward the southeast. Under the remaining climate scenarios, the centroid of *K. pygmaea* shows a consistent northward shift, ranging from 40.74 km to 94.26 km. These findings suggest that, constrained by its growth habits and ecological requirements, the suitable habitat of *K. pygmaea* remains predominantly confined to the high-altitude regions of the Qinghai Tibet Plateau.

## 4. Discussion

### 4.1. Evaluation of the MaxEnt Model

This study employed the MaxEnt model, which is characterized by low sample size requirements and high predictive accuracy, to model the current and future (2050s and 2070s) potential suitable habitats of *K. pygmaea* [11]. To further improve prediction accuracy, the model was optimized to enhance the reliability of the results. MaxEnt outputs showed that under current climatic conditions, the AUC value of the model evaluation for the suitable habitats of *K. pygmaea* with respect to its existing distribution points was greater than 0.9, indicating high predictive accuracy and reliability

### 4.2. Key Environmental Variables Influencing the Distribution of K. pygmaea

*K. pygmaea*, a dominant species in alpine grasslands, exhibits strong adaptability, particularly under cold and humid harsh conditions. This study identifies elevation as a key limiting factor for its distribution, with an optimal elevational range of 3527.99 m to 6054.54 m. This elevational range is primarily located on the Tibetan Plateau, where the environmental conditions, characterized by low annual mean temperature, high solar radiation, short growing season, moderate precipitation, and stable humidity—closely match the physiological and ecological requirements of *K. pygmaea*, providing a suitable habitat for its survival and growth [29]. However, in contrast to previous studies, earlier research has indicated that annual precipitation (Bio12) is a key limiting factor for the potential distribution of *K. pygmaea* [30]. This discrepancy in conclusions may arise from differences in spatial scale, sample distribution, methods of handling variable multicollinearity, and the definition of geographic boundaries [31]. Among all environmental factors, elevation is a comprehensive topographic variable, with environmental elements such as temperature, precipitation, and soil type varying along altitudinal gradients, thus providing strong explanatory power in characterizing species distribution patterns [32]. On the Tibetan Plateau, temperature decreases by approximately 0.6 °C for every 100 m increase in elevation, while precipitation varies significantly due to topographic effects, resulting in distinct dry–wet contrasts and high spatial variability [33]. Elevation integrates environmental gradients and indirectly shapes habitat patterns by modulating precipitation through topography [34].

### 4.3. Response of the Geographical Distribution of K. pygmaea to Future Climate Change

Under current climatic conditions, the suitable habitat of *K. pygmaea* is primarily concentrated in central and southern Tibet, central and southern Qinghai, and northwestern Sichuan, with these three regions collectively forming its core distribution area. The total global suitable habitat area reaches 1.13 × 10^5^ km^2^, accounting for 1.18% of China’s total land area. High-suitability and moderate-suitability areas, measuring 1.76 × 10^4^ km^2^ and 3.2 × 10^4^ km^2^, respectively, together comprise over 44% of the total suitable area, indicating a broad ecological niche and strong environmental adaptability of *K. pygmaea* on the Tibetan Plateau. The high-suitability areas are concentrated in southeastern and southern Tibet, as well as the Aba region of Sichuan, where precipitation is relatively abundant, soils are moist, and anthropogenic disturbances are minimal. In contrast, moderate-suitability areas extend into southern Qinghai, reflecting a spatial gradient of declining habitat quality from the southeast to the northwest. Additionally, except for the Qaidam Basin, where arid conditions, water deficit, and insufficient thermal regimes result in unsuitable habitat, most regions of the Tibetan Plateau still possess the fundamental ecological conditions necessary to support *K. pygmaea*, indicating that the species currently maintains a relatively stable distribution pattern. However, under the context of future climate warming, the suitable habitat of *K. pygmaea* exhibits a dynamic trend of expansion followed by contraction. By the 2050s, the total suitable habitat area is generally projected to be larger than the current extent, particularly under the low-emission scenario SSP126, which shows the most significant expansion. This reflects that, in the short term, climate warming may alleviate thermal constraints in certain alpine regions, thereby broadening the range of suitable habitats for *K. pygmaea.* This adaptive response to climate change may facilitate the range expansion of *K. pygmaea* toward higher elevations and more northern latitudes, particularly in marginal areas where its distribution is currently limited by low-temperature constraints. Similarly, Telwala notes that many plant species in the Himalayas are migrating to higher altitudes due to climate warming, which is consistent with the findings of this study [35]. However, by the 2070s, under continued increases in greenhouse gas concentrations, the total suitable habitat area for *K. pygmaea* is projected to decrease by 9.50% under the SSP370 scenario and by 6.76% under the SSP585 scenario compared to current conditions. The most significant range contractions are primarily concentrated in eastern Sichuan and northern Qinghai. The decline in highly suitable habitat is even more severe, decreasing by 58.75% under SSP370 relative to the present, with major losses occurring in northern Sichuan and southern Tibet. These findings suggest that prolonged climate warming may lead to habitat degradation or even local extirpation, exceeding the species’ adaptive capacity and posing a significant threat to the long-term survival of *K. pygmaea*.

Although this study indicates that the suitable habitat of *K. pygmaea* will remain on the Tibetan Plateau under future climate scenarios, its actual migration capacity depends on biological traits such as seed dispersal rate, soil moisture requirements, and gene flow among populations. Moreover, habitat fragmentation caused by grazing activities and glacial retreat currently constrains its survival [36]. This is consistent with the “migration lag” result proposed by Hoegh-Guldberg et al., in which climate change leads to rapid migration of suitable areas, but slower natural migration of plants [37]. Assisted migration of *K. pygmaea* can help maintain genetic diversity, but its success depends on competition with local species and ecological adaptability [38]. In high-altitude regions, steep terrain and habitat fragmentation often restrict natural species migration, hindering their dispersal to predicted suitable areas [39]. Assisted migration can help species expand into more suitable habitats, reducing the risk of local extinction due to climate warming [40]. Although distribution models provide spatial guidance for future suitable habitats, a comprehensive assessment of *K. pygmaea*’s dispersal potential in fragmented landscapes, and the feasibility of assisted migration, requires integrating landscape connectivity and population genetic studies.

### 4.4. Limitations and Innovations of the Study

Although this study used the MaxEnt model to predict the current and future potential suitable habitats of *K. pygmaea*, certain limitations remain. First, local extreme weather events, land-use changes, and anthropogenic activities were not fully incorporated; these factors may constrain or alter the distribution of suitable habitats for *K. pygmaea.* Additionally, biotic interactions, such as plant–plant interactions, and genetic diversity were not considered in this study. In summary, future research should integrate analyses of extreme climatic events, human disturbances, ecological interactions, and genetic factors to comprehensively assess the responses of *K. pygmaea* to these drivers and the resulting shifts in suitable habitats. Such efforts will provide a robust theoretical foundation for the conservation of dominant species in alpine grasslands and for ecological restoration in these fragile ecosystems.

As the most widely distributed and dominant matrix-forming species in the alpine meadows of the Tibetan Plateau, *K. pygmaea* plays a pivotal role in maintaining community structure and ecosystem functioning. It serves as a foundational species, contributing significantly to soil carbon sequestration, water and soil conservation, and forage provision, key ecosystem services in high-altitude grazing systems. Despite its ecological centrality, there remains a critical lack of systematic and high-resolution modeling efforts to predict its distributional dynamics under future climate change scenarios. This study presents the first comprehensive assessment of the potential habitat shifts of *K. pygmaea* under current and future climate conditions, using high-resolution environmental layers and state-of-the-art species distribution models (SDMs). By moving beyond broad-scale biodiversity patterns or community-level analyses, our approach focuses on the response of a key dominant species to climate forcing, thereby offering novel insights into the mechanisms linking species vulnerability to ecosystem stability. We reveal substantial range contractions, particularly in core suitable areas such as northern Sichuan and southern Tibet, indicating that the loss of this foundation species could trigger cascading effects on alpine grassland integrity, leading to structural degradation and functional decline under ongoing warming.

## 5. Conclusions

This study’s results indicate that elevation (3527.99–6054.54 m) is the dominant factor shaping the habitat suitability of *K. pygmaea.* The core distribution is located in the central and southern Tibetan Plateau, with a current suitable area of 1.13 × 10^5^ km^2^. Under future climate warming, the suitable range is projected to expand initially and then contract, with high-suitability areas declining by nearly 60% by the 2070s under high-emission scenarios (SSP370 and SSP58.5), and the distribution centroid shifting northward. These findings suggest increasing habitat degradation risk for *K. pygmaea* and highlight the high vulnerability of alpine ecosystems to climate change. This study provides a scientific basis for the conservation and adaptive management of the Tibetan Plateau as a critical ecological barrier. Future research should incorporate dispersal limitations and anthropogenic disturbances to improve predictive accuracy.

## Figures and Tables

**Figure 1 plants-14-03585-f001:**
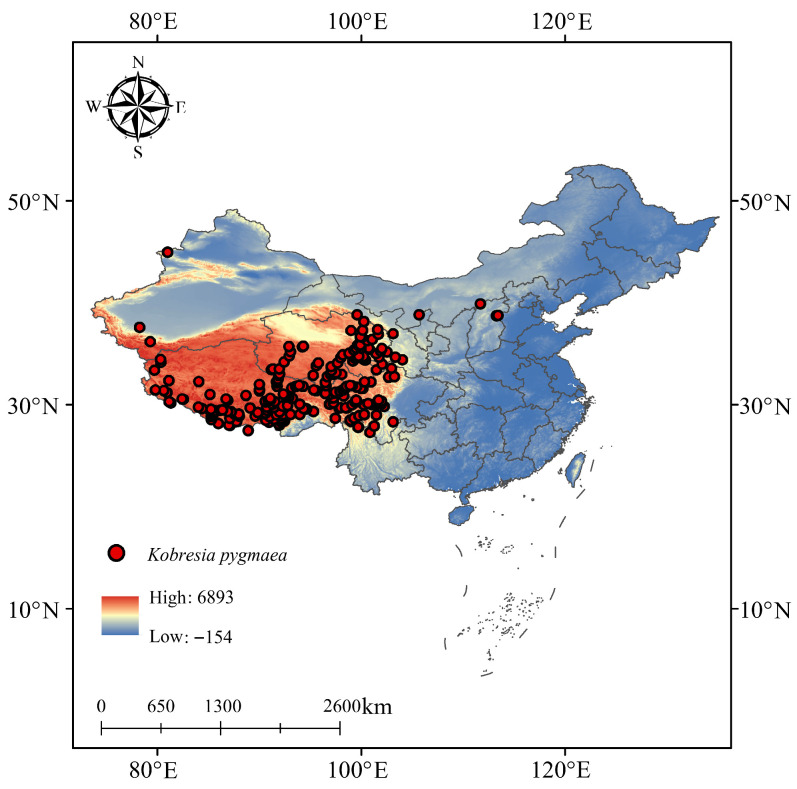
Distribution points of *K. pygmaea* in the world.

**Figure 2 plants-14-03585-f002:**
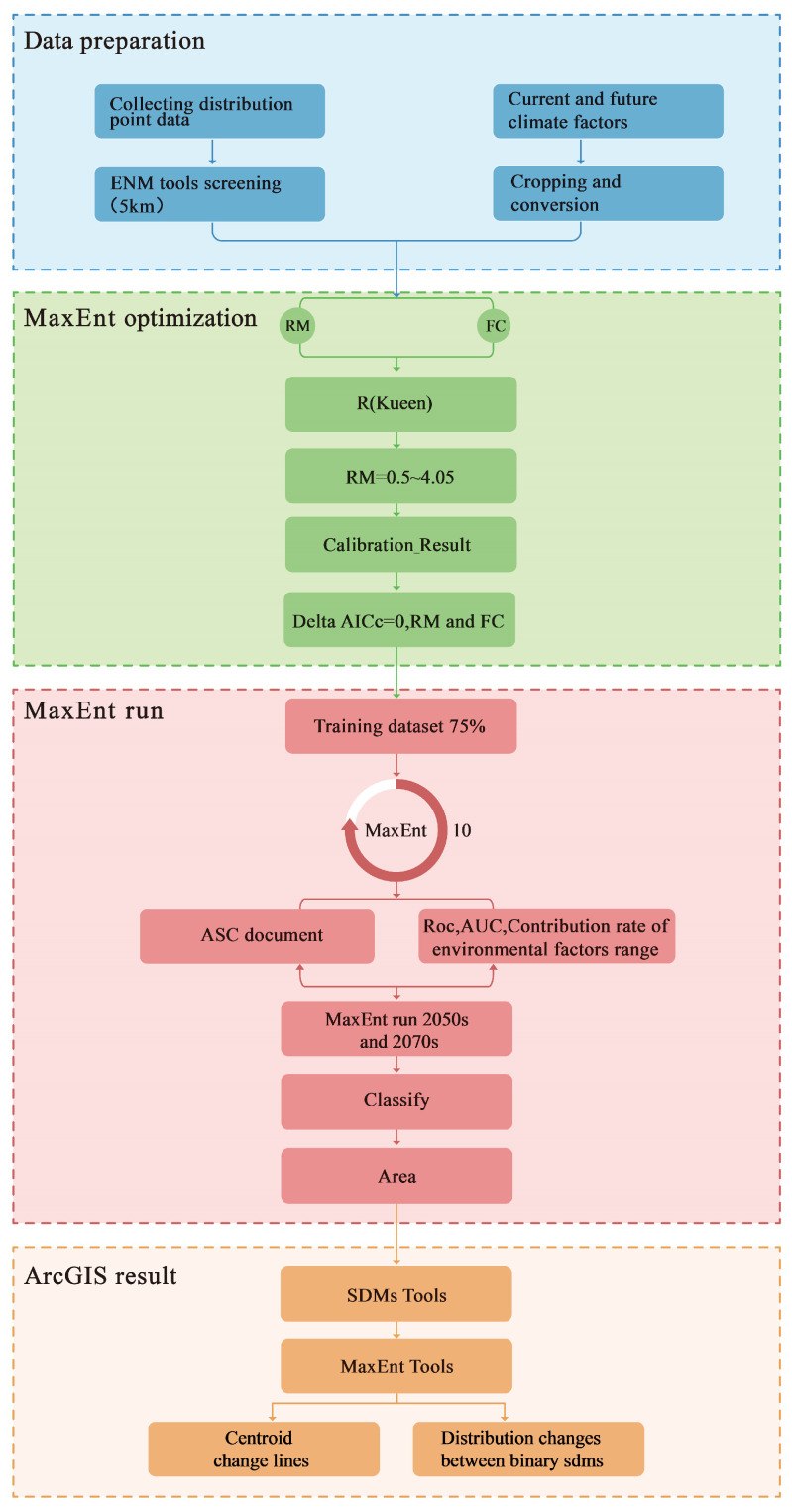
MaxEnt predicts the technical route of suitable areas for *K. pygmaea*.

**Figure 3 plants-14-03585-f003:**
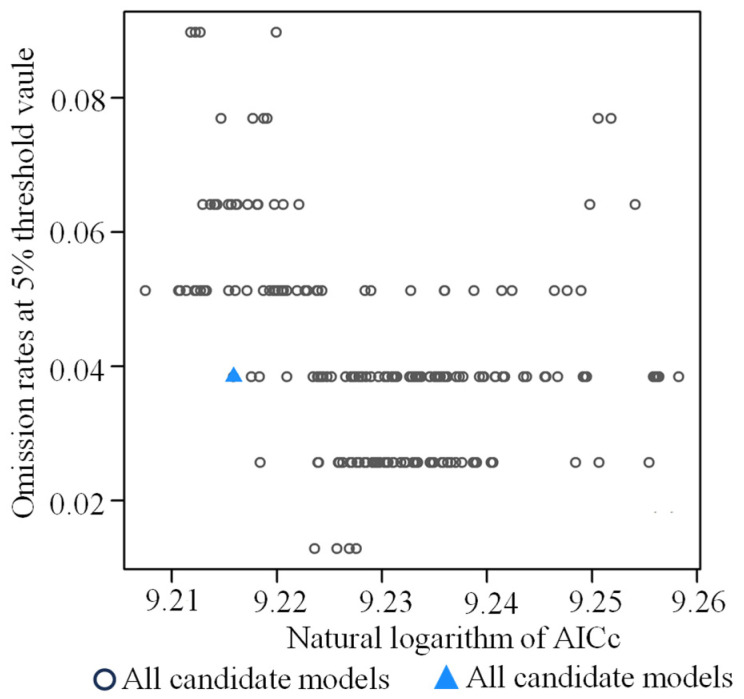
MaxEnt model running parameters calibration.

**Figure 4 plants-14-03585-f004:**
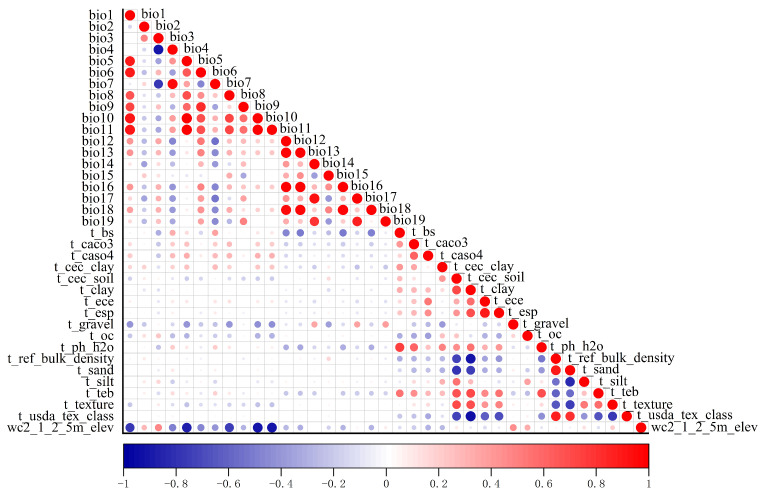
Correlation among environmental variables for *K. pygmaea*.

**Figure 5 plants-14-03585-f005:**
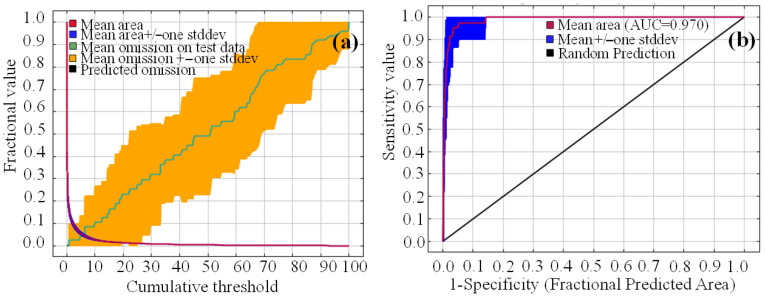
(**a**) Omission rate for accuracy analysis and (**b**) ROC curve of MaxEnt under the current climate conditions.

**Figure 6 plants-14-03585-f006:**
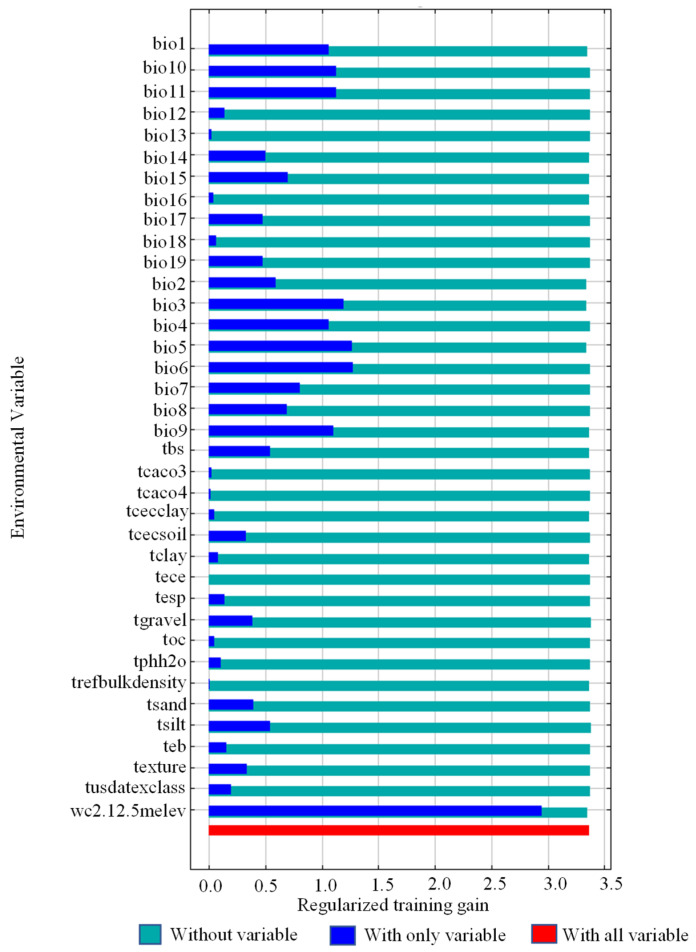
The importance of environmental variables was tested.

**Figure 7 plants-14-03585-f007:**
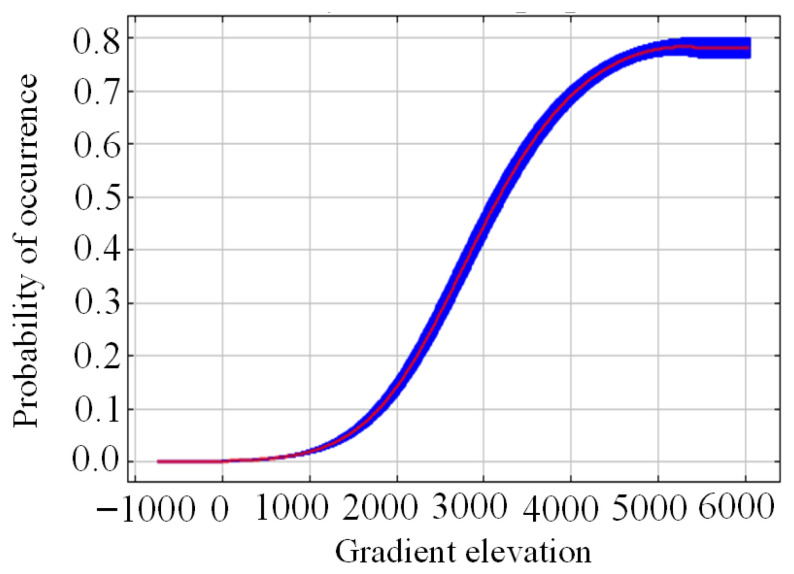
Response curves of dominant environmental variables. Note: The blue area represents the 95% confidence interval.

**Figure 8 plants-14-03585-f008:**
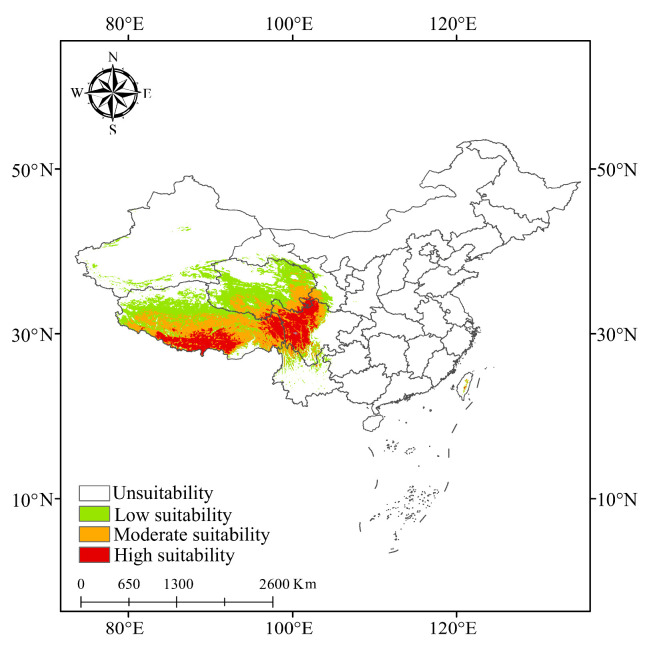
Global suitability zones of *K. pygmaea* under current climate conditions. Note: The figure is based on the standard map No. Gs-jing (2022) 1061 downloaded from the standard map service website of the National Administration of Surveying, Mapping and Geographic Information. The base map is not modified.

**Figure 9 plants-14-03585-f009:**
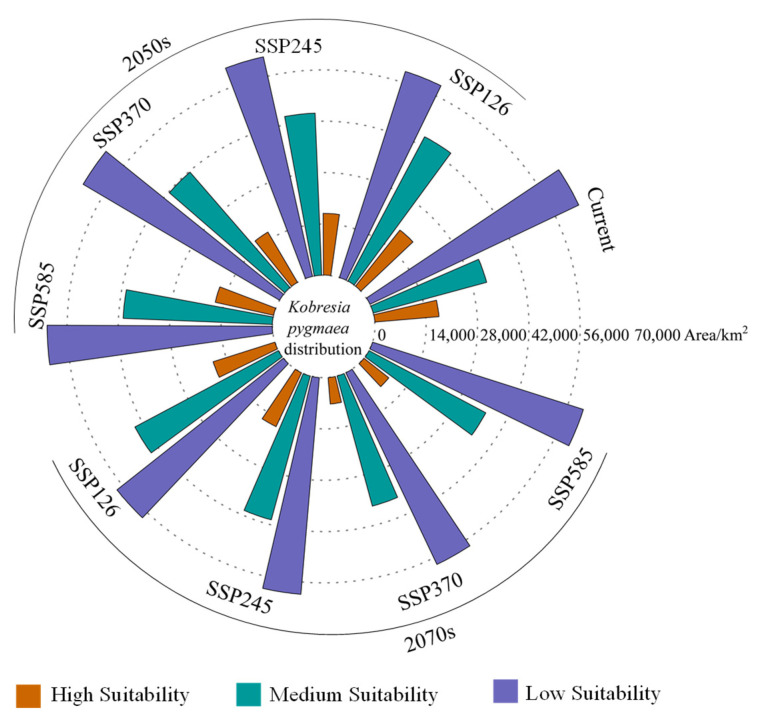
Global habitat area (km^2^) of *K. pygmaea* under different climate scenarios.

**Figure 10 plants-14-03585-f010:**
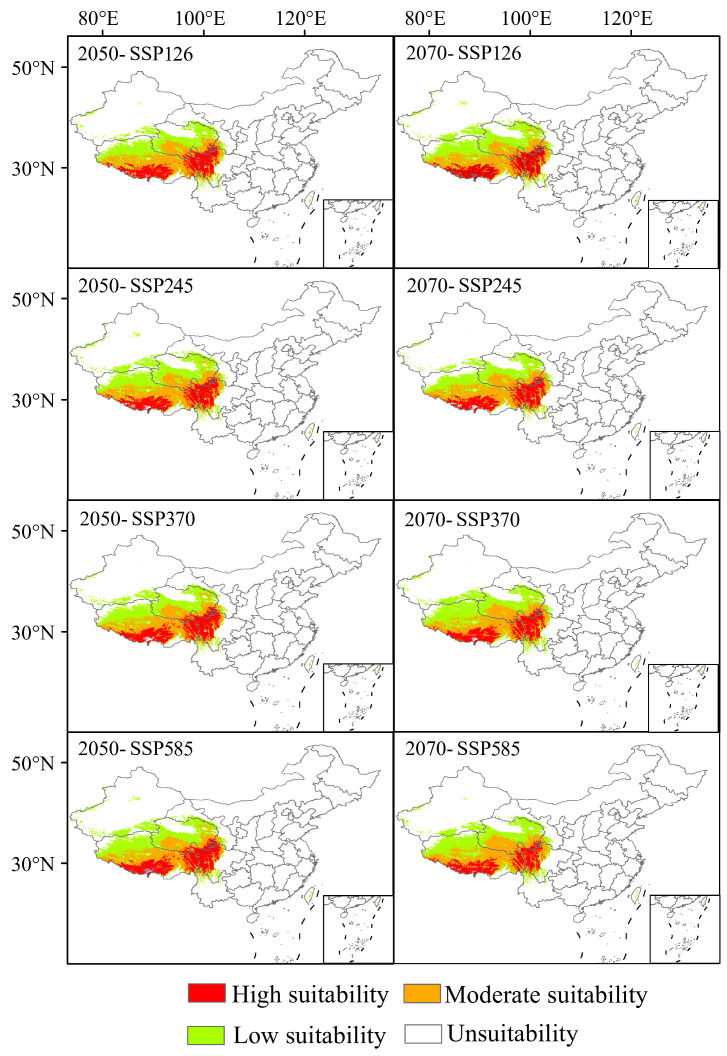
Potential distribution of *K. pygmaea* under different future Climate Scenarios. Note: This map was derived from the unaltered standard map No. Gs-Jing (2022) 1061, which was obtained from the official website of the Ministry of Natural Resources’ standard map service.

**Figure 11 plants-14-03585-f011:**
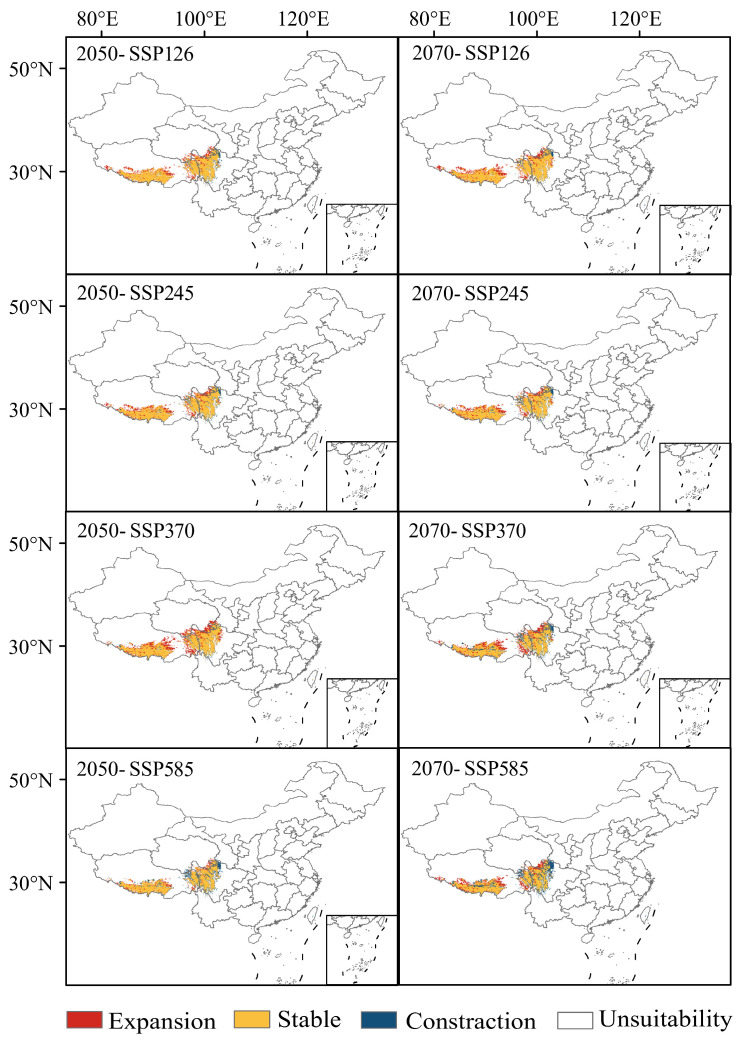
Temporal and spatial evolution trend of *K. pygmaea* distribution under the influence of climate change (Map No.: GS Jing (2022) 1061). Note: This map was derived from the unaltered standard map No. Gs-Jing (2022) 1061, which was obtained from the official website of the Ministry of Natural Resources standard map service.

**Figure 12 plants-14-03585-f012:**
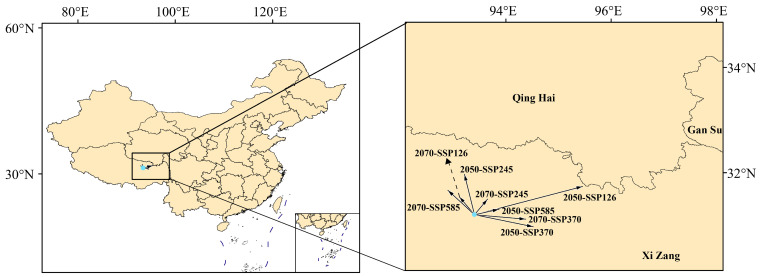
Change of centroid migration of *K. pygmaea* under the influence of climate change. Note: The blue dots represent the origin points.

**Table 1 plants-14-03585-t001:** Environmental variables involved in the MaxEnt model construction and their contribution rates.

Variable	Name	Percent Contribution (%)
wc2.1_2.5m_elev	Altitude	66.4
bio9	Mean temperature of driest quarter (°C)	10.3
bio1	Annual mean temperature	3.1
bio18	Precipitation of the warmest quarter (mm)	2.2
bio14	Precipitation of driest month (mm)	1.4
bio4	Temperature seasonality	1.4
bio6	Minimum temperature of coldest month (°C)	0.9
t_bs	Topsoil base saturation	0.5

## Data Availability

The original contributions presented in this study are included in this article; further inquiries can be directed to the corresponding authors.

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
