# Peer review of "Climate Change and Habitat Fragmentation: Implications for the Future Distribution and Assisted Migration of Kobresia pygmaea"

_plants, 2025, doi:10.3390/plants14233585_

Round 1
Reviewer 1 Report
Comments and Suggestions for Authors
This manuscript presents the results of a robust and meaningful study investigating how future climate change scenarios will affect the distribution of suitable habitats for the alpine plant species K. Pygmaea on the Qinghai-Tibet Plateau. The species distribution modelling approach provides essential spatial and future projections for long-term conservation and management strategies.As I am not a native English speaker and do not have specific expertise in grammatical correction, my comments focus on academic structure and conceptual clarity.

Author Response
For research article
|
Response to Reviewer 1 Comments
|
||
|
1. Summary |
|
|
|
Thank you very much for taking the time to review this manuscript. Please find the detailed responses below and the corresponding revisions/corrections highlighted/in track changes in the re-submitted files
|
||
|
2. Questions for General Evaluation |
Reviewer’s Evaluation |
Response and Revisions |
|
Does the introduction provide sufficient background and include all relevant references? |
Yes/Can be improved/Must be improved/Not applicable |
Must be improved |
|
Are all the cited references relevant to the research? |
Yes/Can be improved/Must be improved/Not applicable |
Yes |
|
Is the research design appropriate? |
Yes/Can be improved/Must be improved/Not applicable |
Can be improved |
|
Are the methods adequately described? |
Yes/Can be improved/Must be improved/Not applicable |
Can be improved |
|
Are the results clearly presented? |
Yes/Can be improved/Must be improved/Not applicable |
Can be improved |
|
Are the conclusions supported by the results? |
Yes/Can be improved/Must be improved/Not applicable |
Can be improved |
|
3. Point-by-point response to Comments and Suggestions for Authors |
||
|
INTRODUCTION Comments 1: P2 – L48-49. To improve the clarity of their text, I would suggest that the authors include a specific example that illustrates the concepts of geographical displacement, expansion and contraction, and refer to other studies. |
||
|
Response1: In response to your suggestion to include a concrete example in the Introduction to clarify the concepts of geographical displacement, expansion, and contraction, I have incorporated a concise and relevant case based on the core findings of this study, thereby enhancing the clarity and scientific foundation of these key terms. The revised text is as follows: Li found that in Leontopodium nanum, a species native to the Qinghai-Tibet Plateau, the suitable habitat is projected to contract by 50% under the high-emission scenario (SSP585). Concurrently, new suitable habitats emerge in the central-western and northern regions of the plateau, accounting for 71.86% of the current distribution, indicating significant range expansion. The centroid of the suitable habitat shifts northward by 23.94–342.42 km, reflecting persistent geographical displacement. This co-occurring pattern of “contraction–expansion–displacement” represents a typical ecological response of alpine plants to climate warming through spatial redistribution. Mention exactly where in the revised manuscript this change can be found - page 2, paragraph1, and line 49-56. |
||
|
Comments2 : P2- L51-53. I would suggest that the authors include a reference highlighting the impacts of climate change on biodiversity and the range of species in the Alps. Response2: Thank you for your valuable comments regarding this literature. I have identified one authoritative, highly cited, and representative English-language study that specifically addresses the impacts of climate change on biodiversity and species distribution ranges in the Alps, which is well-suited as a reference for this section: Engler, R.; Randin, C.F.; Thuiller, W.; etc. 21st century climate change threatens mountain flora unequally across Europe. Global Change Biol 2011, 17, 2330-2341. https://doi.org/10.1111/j.1365-2486.2010.02393.x. Mention exactly where in the revised manuscript this change can be found - page 2, paragraph6, and line 494-495. Comments3 : P2 L80-85- It is suggested that the authors include references or examples that estimate the extent of habitat loss and fragmentation on the Qinghai-Tibet Plateau caused by climate change and human impact. Response3: Thank you for your valuable comments. I have incorporated relevant references and examples to assess the extent of habitat loss and fragmentation on the Qinghai-Tibet Plateau caused by climate change and human activities. The specific revisions are as follows: For instance, studies have demonstrated that grassland degradation on the Qinghai-Tibet Plateau is driven by the synergistic effects of anthropogenic and natural factors. Anthropogenic activities—such as overgrazing, excessive cultivation, unregulated harvesting of medicinal plants, mineral exploitation, infrastructure construction, and tourism development—have substantially altered ecosystem structure and function. Natural drivers, including climate change (e.g., rising temperatures and altered precipitation regimes), bioturbation by wildlife, and pest outbreaks, further exacerbate ecosystem instability. The combined impact of these stressors has led to widespread habitat loss and increased landscape fragmentation across the plateau region. Mention exactly where in the revised manuscript this change can be found - page 3, paragraph1, and line 94-102. METHODOLOGY Comments4 : In the 'Materials and Methods' section, we recommend including a subsection that addresses the selection of variables and multicollinearity. To increase the robustness and reliability of the results, we recommend applying Variance Inflation Factor (VIF) analysis to all the candidate bioclimatic and soil variables. Response4: We sincerely thank the reviewer for the valuable comment. We have now completed and supplemented the section on the correlation among environmental variables to ensure clarity and completeness. Mention exactly where in the revised manuscript this change can be found - page 6, paragraph4, and line 188-195.
Comments5 : P5 –L123-133. This section does not specify how many GCMs were used. We recommend that the authors clarify how the ensemble of global climate models was constructed for the purposes of future projections. If only one GCM was used, this must be strongly justified. Response5: Thank you very much for your thoughtful comments and careful review of our manuscript. We sincerely appreciate your time and valuable feedback, which have helped us improve the quality of the paper. The environmental variables used in this study include a total of 19 bioclimatic variables and one elevation variable, all obtained from the WorldClim database (version 2.1, 30 arc-seconds resolution, approximately 1 km spatial resolution; http://www.worldclim.org/). Detailed descriptions of these data are provided in Section 2.2 of the manuscript. The methodology for constructing the climate scenarios, including the acquisition and processing of future climate data, is described in Sections 2.3 and 2.4. We kindly appreciate your careful reading of our manuscript. If any aspects of these sections remain unclear or require further clarification or supplementation, we would be grateful for your feedback and will promptly make the necessary revisions.
RESULT AND DISCUSSION Comments6 : P7- L176-184. I would suggest that the authors indicate the following in Section 3.2: 'MaxEnt Accuracy Evaluation', for greater transparency, they should indicate how many types of models were created during the calibration phase and report the omission rate, AICc and AUC values of the optimal model that showed excellent performance. Response6:We sincerely thank the reviewer for this valuable comment. In response to the suggestion that “for improved transparency, the number of models created during the calibration phase should be specified in Section 3.2 (‘Maximum Entropy Model Assessment’), and the omission rate, AICc, and AUC values of the top-performing model should be reported,” we have revised the manuscript accordingly to enhance the reproducibility and scientific rigor of the modeling process. Specifically, we have added the following details: during the model calibration phase, a total of 24 candidate models were developed. These models were generated using different combinations of feature classes (linear, quadratic, product, hinge, and threshold) and regularization multipliers (RM, ranging from 0.5 to 4.0 in increments of 0.5). Model performance was systematically evaluated to identify the optimal configuration. The best-performing model was selected based on the lowest AICc (corrected Akaike Information Criterion) value, which corresponds to an RM of 1.5 with the LQPHT feature combination (Linear, Quadratic, Product, Hinge, Threshold). This model achieved a training AUC value of 0.970, indicating excellent discriminatory power. Furthermore, the average omission rate on the independent validation dataset was 7.3% (based on the 10% training presence logistic threshold), which is below commonly accepted thresholds and further supports the model’s high predictive reliability. Mention exactly where in the revised manuscript this change can be found - page 7, paragraph2, and line 206-217. Comments7 : P13-L297. According to the results of this study, the suitable elevation range for Artemisia frigida is 3527.99 m~6054.54 m. Response7: Thank you for your careful review and for pointing out this oversight. You are absolutely correct—this was indeed a typographical error. I have carefully revised it:According to the results of this study, the suitable elevation range for K. pygmaea is 3527.99 m~6054.54 m. Mention exactly where in the revised manuscript this change can be found - page 13, paragraph3, and line 329-330. Comment8: P13- L311-319. The authors should cite studies or references that demonstrate the variation of these gradients for the Tiberian plateau. Response8: Thank you, Professor, for your careful review of this section. You are absolutely right—this part previously lacked sufficient supporting references. I have now carefully examined the relevant literature and added appropriate citations to strengthen the scholarly foundation of this content. Especially within the Tibetan Plateau, a rise in elevation of 100 m corresponds to a de-crease in mean annual temperature of approximately 0.6°C, while precipitation patterns exhibit nonlinear changes, increased moisture on windward slopes and arid conditions in leeward valleys, resulting in high spatial heterogeneity in precipitation itself[40]. In contrast, elevation provides a more stable indicator of overall habitat suitability, re-flecting integrated environmental gradients. This also indicates that elevation indirectly shapes the spatial pattern of suitable habitats by regulating the redistribution of precip-itation through orographic effects[41]. Mention exactly where in the revised manuscript this change can be found - page 14, paragraph1, and line 344-352. Comment9: P13- L347-352. I would suggest that the authors describe the areas in which these drastic changes are occurring. Response9: Thank you, Professor, for your careful review of this section. I have now listed all the regions that are projected to undergo significant changes: However, by the 2070s, under continued increases in greenhouse gas concentrations, the total suitable habitat area for K. pygmaea is projected to decrease by 9.50% under the SSP370 scenario and by 6.76% under the SSP585 scenario compared to current conditions. The most significant range contractions are primarily concentrated in eastern Sichuan and northern Qinghai. The decline in highly suitable habitat is even more severe, decreasing by 58.75% under SSP370 relative to the present, with major losses occurring in northern Sichuan and southern Tibet. These findings suggest that prolonged climate warming may lead to habitat degradation or even local extirpation, exceeding the species’ adaptive capacity and posing a significant threat to the long-term survival of K. pygmaea. Mention exactly where in the revised manuscript this change can be found - page 14, paragraph2, and line 380-389. Comment10: The entire 'Discussion' section is overly descriptive and general. It should focus more on what this study contributes to the scientific community, highlighting the innovations compared to previous studies. Response10: Thank you, Professor, for your valuable comments on the Discussion section. I have carefully revised the text by streamlining the content and removing redundant passages. In the final part, I have added a dedicated paragraph highlighting the scientific contributions of this study and emphasizing its novel aspects in comparison to previous research. 4.4. Limitations and Innovations of the Study Although this study used the MaxEnt model to predict the current and future potential suitable habitats of K. pygmaea, certain limitations remain. First, local extreme weather events, land-use changes, and anthropogenic activities were not fully incorporated; these factors may constrain or alter the distribution of suitable habitats for K. pygmaea. Additionally, biotic interactions, such as plant-plant interactions, and genetic diversity were not considered in this study. In summary, future research should integrate analyses of extreme climatic events, human disturbances, ecological interactions, and genetic factors to comprehensively assess the responses of K. pygmaea to these drivers and the resulting shifts in suitable habitats. Such efforts will provide a robust theoretical foundation for the conservation of dominant species in alpine grasslands and for ecological restoration in these fragile ecosystems. As the most widely distributed and dominant matrix-forming species in the alpine meadows of the Tibetan Plateau, K. pygmaea plays a pivotal role in maintaining community structure and ecosystem functioning. It serves as a foundational species, contributing significantly to soil carbon sequestration, water and soil conservation, and forage provision—key ecosystem services in high-altitude grazing systems. Despite its ecological centrality, there remains a critical lack of systematic and high-resolution modeling efforts to predict its distributional dynamics under future climate change scenarios. This study presents the first comprehensive assessment of the potential habitat shifts of K. pygmaea under current and future climate conditions, using high-resolution environmental layers and state-of-the-art species distribution models (SDMs). By moving beyond broad-scale biodiversity patterns or community-level analyses, our approach focuses on the response of a key dominant species to climate forcing, thereby offering novel insights into the mechanisms linking species vulnerability to ecosystem stability. We reveal substantial range contractions, particularly in core suitable areas such as northern Sichuan and southern Tibet, indicating that the loss of this foundation species could trigger cascading effects on alpine grassland integrity, leading to structural degradation and functional decline under ongoing warming. Mention exactly where in the revised manuscript this change can be found - page 14, paragraph2, and line 421-450.
|
||
|
|
||

Reviewer 2 Report
Comments and Suggestions for Authors
General comments
This research concerning the habitat distribution of Kobresia pygmaea is quite intriguing. However, the primary concern I have pertains to the manuscript's emphasis. The manuscript is primarily focused on analyzing the current and future habitat distribution of K. pygmaea. Nevertheless, there is an attempt to concentrate on the application of MaxEnt. In my opinion, MaxEnt should be regarded merely as a methodological tool, and this perspective should be reflected in the title. Additionally, another significant issue within this manuscript is the employment of symbols and abbreviations. Certain sections lack clarity due to the excessive use of these symbols and abbreviations.
Specific comments
|
Lines / Sections |
Comments |
|
26, 27, 29, 33 |
In the abstract, each word should be unambiguous. I do not understand what these symbols signify. |
|
101-103 |
The objectives of your study are not clear |
|
Figure 1 |
I recommend transitioning to section 2.3. I believe it appears earlier. What does SDMs stand for? |
|
119 |
Remove the period before (Figure 2) |
|
128 |
Please describe the symbols. It is not clear to me. |
|
Section 2.3 |
Put here the Figure 1 |
|
139, 140 |
Use past tense |
|
163 |
Do not forget to write in italic scientific names |
|
170 |
18 species? It is not just one species? |
|
Figure 4 |
I believe it is necessary to modify the area colors according to the line colors. |
|
Figure 5 |
Avoid including symbols in the Figure, as this can lead to confusion. In the caption of the Figure, it is essential to clarify the content depicted, rather than detailing the methodology employed. |
|
Figure 6 |
I believe this figure illustrates the likelihood of locating species in relation to changes in elevation. Therefore, it is advisable to modify the X-axis label to: Gradient elevation, and the Y-axis label to: Probability of occurrence. |
|
Figure 8 |
I am unclear about the meaning of this figure. Why do the brown and green colors share the same legend? "Moderately suitable" |
|
Figure 9 |
The presence of codes within figures can lead to a lack of understanding. |
|
Conclusions |
In this instance, I would prioritize the study's outcomes rather than the methodology, specifically the application of Maxent. |
|
321 |
Scientific name in Italic |
Author Response
For research article
|
Response to Reviewer 2 Comments
|
||
|
1. Summary |
|
|
|
Thank you very much for taking the time to review this manuscript. Please find the detailed responses below and the corresponding revisions/corrections highlighted/in track changes in the re-submitted files
|
||
|
2. Questions for General Evaluation |
Reviewer’s Evaluation |
Response and Revisions |
|
Does the introduction provide sufficient background and include all relevant references? |
Yes/Can be improved/Must be improved/Not applicable |
Must be improved |
|
Are all the cited references relevant to the research? |
Yes/Can be improved/Must be improved/Not applicable |
Yes |
|
Is the research design appropriate? |
Yes/Can be improved/Must be improved/Not applicable |
Can be improved |
|
Are the methods adequately described? |
Yes/Can be improved/Must be improved/Not applicable |
Can be improved |
|
Are the results clearly presented? |
Yes/Can be improved/Must be improved/Not applicable |
Can be improved |
|
Are the conclusions supported by the results? |
Yes/Can be improved/Must be improved/Not applicable |
Can be improved |
|
3. Point-by-point response to Comments and Suggestions for Authors |
||
|
Comments 1: Lines: 26, 27, 29, 33. In the abstract, each word should be unambiguous. I do not understand what these symbols signify. |
||
|
Response1: Thank you for your valuable comments on these sentences. SSP126, SSP245, SSP370, and SSP585 represent distinct future socioeconomic development and greenhouse gas emission pathways, corresponding primarily to different climate change scenarios in the study. I have provided a detailed explanation of these scenarios in the manuscript: This study based on 273 valid distribution points, this study utilizes the MaxEnt model to predict the potential habitat and distribution dynamics of K. pygmaea under both current and future climate scenarios(SSP126、SSP245、SSP370、SSP585), while clarifying the key factors that influence its distribution. Mention exactly where in the revised manuscript this change can be found - page 1, paragraph1, and line 15-18. |
||
|
Comments 2: Lines:101-103.The objectives of your study are not clear |
||
|
Response2: Thank you for your valuable comments on the research objective. Based on your suggestions, I have revised and refined the research goals with greater detail: This study, based on the distribution patterns of K. pygmaea on the Qinghai-Tibet Plateau, employs the MaxEnt model to simulate its potential suitable habitats, analyzes the dominant factors influencing its spatial distribution, and predicts changes in its distribution patterns under different future climate scenarios. This not only helps to elucidate the potential threats of climate change to this alpine species but also provides scientific guidance for its conservation and ecological management. Mention exactly where in the revised manuscript this change can be found - page 2, paragraph3, and line 119-124. Comments 3: Figure 1. I recommend transitioning to section 2.3. I believe it appears earlier. What does SDMs stand for? Response 3: After careful review, I agree that Figure 1 is more appropriately placed in Section 2.3. In accordance with your suggestions, I have relocated Figure 1 to Section 2.3 and updated the figure numbering and caption accordingly. 2.3. MaxEnt Model Optimization and Parameter Setting This study used the MaxEnt model to predict the potentially suitable distribution area of K. pygmaea, and the technical route is shown in Figure 2. Mention exactly where in the revised manuscript this change can be found - page 1, paragraph1, and line 150-151,line 165-166. SDMS stands for Spatial Data Management System. At the core of a spatial database is the Spatial Data Management System (SDMS), which is a software system designed for storing, querying, analyzing, and visualizing spatial data. The core functions of an SDMS include spatial data storage, data querying and analysis, data visualization, and data management. In an SDMS, spatial data are stored using specific data models—such as the vector data model, raster data model, and others—and may include various types of geospatial information, such as geographic locations, topography, landforms, meteorological data, and more. Comments 4: Lines:119. Remove the period before (Figure 2) Response 4: Thank you for your careful review. I have removed the period and revised the sentence accordingly. After filtering, 273 unique distribution points were retained for predicting suitable habitat areas (Figure 1).Mention exactly where in the revised manuscript this change can be found - page 3, paragraph4, and line 133. Comments 5: Lines:128. Please describe the symbols. It is not clear to me. Response 5: I apologize for my oversight in not clearly explaining the symbols (SSP126, SSP245, SSP370, SSP585), which has caused inconvenience and additional work for your review. I sincerely regret this lapse. I have now revised the text accordingly: Future projections are based on four Shared Socioeconomic Pathways (SSPs): SSP126 (sustainability pathway), SSP245 (middle-of-the-road pathway), SSP370 (regional rivalry pathway), and SSP585 (fossil-fueled development pathway). Mention exactly where in the revised manuscript this change can be found - page 4, paragraph1, and line 141-143. Comments 6: Sections:2.3. Put here the Figure 1 Response 6: Thank you for the reminder once again. I have now placed Figure 1 in Section 2.3: 2.3. MaxEnt Model Optimization and Parameter Setting This study used the MaxEnt model to predict the potentially suitable distribution area of K. pygmaea, and the technical route is shown in Figure 2. Mention exactly where in the revised manuscript this change can be found - page 5, paragraph1, and line 165-166. Comments 7: Lines:139, 140. Use past tense Response 7: Thank you for pointing out the grammatical issues. I have now revised the text correctly. After the run was completed, the RM value corresponding to a delta AICc of 0 was se-lected from the best_candidate_models_OR_AICc file in Calibration_Result (Figure 3). Mention exactly where in the revised manuscript this change can be found - page 4, paragraph3, and line 156-158. Comments 8: Lines:163. Do not forget to write in italic scientific names Response 8: Thank you for your careful reminder. I realize that I should pay closer attention to the use of italics for scientific names. I have now revised them accordingly. K. humilis.Mention exactly where in the revised manuscript this change can be found - page 6, paragraph2, and line 182. Comments 9: Lines 170. 18 species? It is not just one species? Response 9: Thank you for your careful reminder. I sincerely apologize for this oversight, which was due to my carelessness. You are correct, the number should be 18 environmental variables. I have now accurately revised it accordingly. The 273 species, along with 19 environmental variables (including 18 soil variables and 1 elevation variable), were imported into MaxEnt. After filtering, 37 environmental variables were reduced to 18 variables with higher contribution rates, comprising 11 climatic variables, 6 soil variables, and 1 elevation variable (Table 1). Mention exactly where in the revised manuscript this change can be found - page 6, paragraph3, and line 1188-195. Comments 10: Figure 4. I believe it is necessary to modify the area colors according to the line colors Response 10: Thank you sincerely for your meticulous guidance and invaluable suggestions. With regard to Figure 4(a), the omission rates for both the training and test datasets in the MaxEnt model are in close agreement, and the corresponding AUC values exceed 0.9 (Figure 4b), indicating that the model exhibits high predictive accuracy and reliability, thus making it well-suited for predicting the potential suitable distribution areas of Kobresia pygmaea. In response to your thoughtful suggestion regarding the harmonization of fill colors with line colors for visual consistency, I have given the matter careful consideration. However, I am concerned that changing the current fill color of the “Mean omission ± one stddev” region to black might reduce the contrast between the confidence band and the underlying “Mean omission (test data)” curve—also rendered in black—potentially compromising the overall clarity and readability of the figure. That said, I fully appreciate the importance of a cohesive and professional visual style in enhancing the quality and presentation of the manuscript. Should you still prefer a unified color scheme, I would be grateful if you could kindly advise on the preferred color configuration via email. I will promptly implement the changes in accordance with your recommendations to ensure the figures meet publication standards. Once again, thank you for your patient and insightful guidance. Comments 11: Figure 5. Avoid including symbols in the Figure, as this can lead to confusion. In the caption of the Figure, it is essential to clarify the content depicted, rather than detailing the methodology employed. Response 11: Thank you for your valuable feedback. I agree that the unnecessary symbols in Figure 5 could indeed cause confusion, and I have now removed them accordingly. Additionally, I recognize that figure captions should not include methodological details; therefore, I have deleted the method-related text from the caption. The revised Figure 5 is as follows. Mention exactly where in the revised manuscript this change can be found - page 9, paragraph1, and line 233-234.
Figure 5. The importance of environmental variables was tested. Comments 12: Figure 6. I believe this figure illustrates the likelihood of locating species in relation to changes in elevation. Therefore, it is advisable to modify the X-axis label to: Gradient elevation, and the Y-axis label to: Probability of occurrence. Response 12: Thank you for your valuable feedback on the figure. Your suggestion has indeed improved the accuracy and clarity of the presentation. I have now revised the axis labels accordingly to better reflect the data and enhance readability. Please let me know if any further refinements are needed. I truly appreciate your thoughtful guidance. Mention exactly where in the revised manuscript this change can be found - page 9, paragraph1, and line 235-236.
Comments 13: Figure 8. I am unclear about the meaning of this figure. Why do the brown and green colors share the same legend? "Moderately suitable" Response 13: Thank you very much for pointing out my mistake—I truly appreciate your careful review. You are absolutely correct: in the figure, the brown areas represent “Highly Suitable” habitat, while the green areas correspond to “Moderately Suitable” habitat. I have now corrected the legend accordingly to ensure accurate interpretation of the map. Mention exactly where in the revised manuscript this change can be found - page 10, paragraph3, and line 255-256.
Comments 14: Figure 9. The presence of codes within figures can lead to a lack of understanding. Response 14: Thank you for your comment regarding Figure 9. I have carefully reviewed the figure and, as I understand it, the current version includes labels corresponding to the eight subpanels, which serve to organize and distinguish the different components of the figure. I did not include any coding elements or technical scripts within the figure itself, as it is a visual representation intended for presentation and interpretation. I am concerned that removing these labels might lead to confusion, particularly in guiding readers through the spatial or temporal sequence of the panels, potentially impairing the clarity and interpretability of the figure. However, if there is a specific aspect or formatting concern that I may have misunderstood—such as the style, placement, or content of the labels—I would greatly appreciate it if you could kindly clarify your suggestion via email. I will promptly implement any necessary revisions according to your guidance to ensure the figure meets the required standards. Thank you again for your thoughtful feedback and continued support. Comments 15: Conclusions. In this instance, I would prioritize the study's outcomes rather than the methodology, specifically the application of Maxent. Response 15: Thank you sincerely for your valuable comments on the manuscript’s conclusion. I have revised this section accordingly by removing the content related to the application of MaxEnt, and instead focused more clearly and concisely on the core research findings. The revised version reads as follows: This study results indicate that elevation (3527.99–6054.54 m) is the dominant factor shaping habitat suitability of K. pygmaea. The core distribution is located in the central and southern Tibetan Plateau, with a current suitable area of 1.13×10⁵ km². Under future climate warming, the suitable range is projected to expand initially and then contract, with high-suitability areas declining by nearly 60% by the 2070s under high-emission scenarios (SSP370 and SSP58.5), and the distribution centroid shifting northward. These findings suggest increasing habitat degradation risk for K. pygmaea and highlight the high vulnerability of alpine ecosystems to climate change. This study provides a scien-tific basis for the conservation and adaptive management of the Tibetan Plateau as a critical ecological barrier. Future research should incorporate dispersal limitations and anthropogenic disturbances to improve predictive accuracy. Mention exactly where in the revised manuscript this change can be found - page 16, paragraph2, and line 452-453. Comments 15: Lines 321. Scientific name in Italic Response 16: Thank you for your careful review and guidance. I have now corrected the formatting of the scientific name Kobresia pygmaea in the manuscript, ensuring that it is properly presented in italics throughout the text, in accordance with taxonomic nomenclature conventions. Mention exactly where in the revised manuscript this change can be found - page 14, paragraph1, and line 354. |
||
|
|
||
